# Learning Dynamic Representations for Discourse Dependency Parsing

**Tianyi Liu**[1,2] and **Yansong Feng**[1*] and **Dongyan Zhao**[1,2]
[1]Wangxuan Institute of Computer Technology, Peking University, China
[2]Center for Data Science, Peking University, China
{liu-tian-yi,fengyansong,zhaody}@pku.edu.cn

## Abstract

Transition systems have been widely used for the discourse dependency parsing task. Existing works often characterize transition states by examining a certain number of elementary discourse units (EDUs), while neglecting the arcs obtained from the transition history. In this paper, we propose to employ GAT-based encoder to learn dynamic representations for sub-trees constructed in previous transition steps. By incorporating these representations, our model is able to retain accessibility to all parsed EDUs through the obtained arcs, thus better utilizing the structural information of the document, particularly when handling lengthy text spans with complex structures. For the discourse relation recognition task, we employ edge-featured GATs to derive better representations for EDU pairs. Experimental results show that our model can achieve state-of-the-art performance on widely adopted datasets including RST-DT, SciDTB and CDTB. Our code is available at https://github.com/lty-lty/Discourse-Dependency-Parsing.

## 1 Introduction

Discourse parsing is the task to study the inner structure of documents by analysing the relationship between text spans known as elementary discourse units (EDUs). It is an important research topic in natural language processing and benefits many downstream tasks, including sentiment analysis (Bhatia et al., 2015), text categorization (Ji and Smith, 2017), summarization (Xu et al., 2020) and so on.

Existing methods for discourse parsing include transition-based models (Jia et al., 2018; Yu et al., 2018; Hung et al., 2020) and graph-based models (Li et al., 2014a). Recent works also employ top-down models (Koto et al., 2021; Zhang et al., 2020; Zhang et al., 2021b). However, their results are not

---

*Corresponding author.

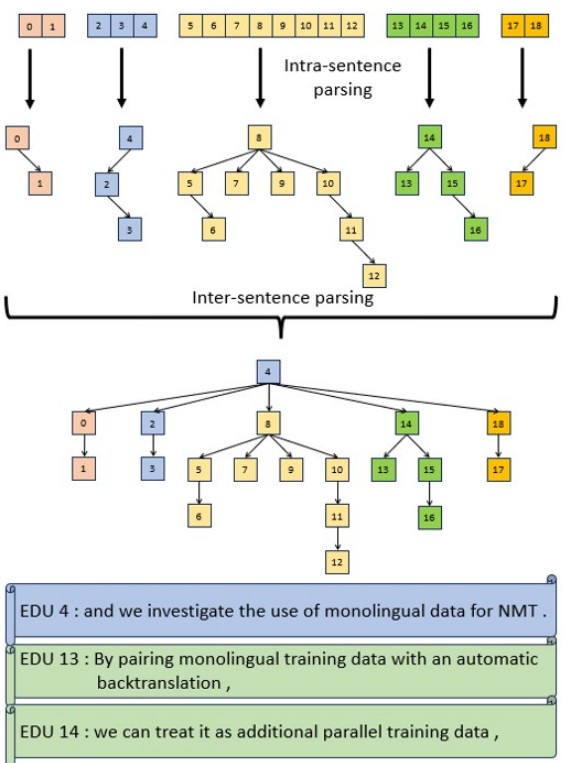

Figure 1: The hierarchical parsing process for structure prediction. Sub-trees are built for each sentence and subsequently merged into a complete dependency tree for the entire document.

satisfactory when compared with syntactic parsing, mainly attributed to the difficulty of encoding EDUs properly and transferring information over long text spans.

Utilizing hierarchical models is a natural way to alleviate the problem. Figure 1 shows the hierarchical parsing process of a document with 5 sentences, a total of 19 EDUs. If we parse the document in a normal manner, it is challenging to connect EDU 4 with EDU 18 correctly due to their considerable distance. But it is much easier when we parse the document at intra-sentence level and inter-sentence level, respectively.

Recent research has demonstrated the advantages of hierarchical discourse parsers across transition-based, graph-based, and top-down models. Kobayashi et al. (2020) utilizes a top-down parsing approach with three levels of granularity: paragraphs in each document, sentences in each paragraph, and EDUs in each sentence. Zhang et al. (2021a) adopts Eisner's algorithm (Eisner, 1996) for structure and relation prediction at intra-sentence, intra-paragraph and intra-document levels. Zhou and Feng (2022) proposes a hierarchical transition model to predict the dependency tree structure at intra-sentence and inter-sentence levels and employs a stacked LSTM (Hochreiter and Schmidhuber, 1997) model to predict the relations.

The previous transition-based and top-down models, despite benefiting from hierarchical structures, suffer from a common issue: they do not effectively utilize lower-level structures for upper-level parsing. Kobayashi et al. (2020) represents a span as a whole by employing a weighted sum over word representations. Zhang et al. (2021a) only assigns complete and incomplete scores to each span. Zhou and Feng (2022) focuses exclusively on root EDUs of sentences for inter-sentence parsing. These approaches fail to leverage the structural information acquired from preceding lower-level parsing stages during the upper-level parsing process. This may hurt the model performance. For example, during inter-sentence parsing in Figure 1, the key information in EDU 13 (*monolingual, backtranslation*) is crucial when we consider the relationship between EDU 4 and EDU 14. Therefore, it is important for us to leverage the parsing history in subsequent steps.

To address this issue, we propose a solution in the form of a hierarchical transition-based model. Our approach mitigates the aforementioned problem by employing graph attention networks (GATs Veličković et al., 2018) to extract features for the sub-trees constructed in the preceding parsing steps. In contrast to Zhou and Feng (2022), where an EDU becomes inaccessible once it is popped from the stack, our method maintains accessibility to it through the obtained arcs, enabling its reuse in subsequent parsing procedures. For example, when we consider the relationship between EDU 4 and EDU 8, we can use information from both intra-sentence parsing history (arc $4 \to 2$) and parsing history of the current inter-sentence sequence (arc $4 \to 0$), instead of relying solely on the isolated

root EDU of the sentence. Furthermore, we improve the performance of discourse relation recognition by employing edge-featured graph attention networks (EGATs; Chen and Chen, 2021). When compared to existing sequence labeling models, EGATs outperform in leveraging precise structure information and predicting relations between distant EDU pairs. Experimental results demonstrate that by utilizing more contextual information with GATs, our models achieve state-of-the-art performance on RST-DT (Carlson et al., 2001), SciDTB (Yang and Li, 2018) and CDTB (Li et al., 2014b) for both structure prediction and relation recognition.

Our main contributions are as follows: (1) We propose a new transition-based model that incorporates a GAT-based sub-tree encoder, effectively leveraging the parsing history. (2) We evaluate two different types of graphs and demonstrate that besides EDUs, the structure information also matters in predicting head-dependant relations, especially when dealing with long text spans with complex structures. (3) We apply EGATs, instead of sequence labeling models, to discourse relation recognition to better leverage structure information. (4) Our model outperforms existing methods by a large margin in popular discourse parsing datasets of different languages and genres.

## 2  Related work

**Pre-trained Language Models**  Recent works have shown that pre-trained language models(PLMs) have achieved good results in discourse parsing (Koto et al., 2021; Nguyen et al., 2021; Hung et al., 2020). These works prove that pre-trained language models can better capture relationship between EDUs than hand-crafted features. Yu et al. (2022) points out that PLMs are trained with sentence-level contexts, rather than the EDU-level structure in discourse parsing. They perform second-stage EDU-level pre-training with two tasks, next EDU prediction and discourse marker prediction, and these adaptations achieve great improvements. For fair comparison, our model and the baselines in this paper use the same PLMs: *bert-base-uncased* for English and *bert-base-chinese* for Chinese.

**Discourse Dependency Parsing**  Compared to discourse constituency parsing, discourse dependency parsing receives relatively less attention. Li et al. (2014a) converts constituency trees in RST-

DT to dependency trees, and employ two graph models on the new dataset. Yang and Li (2018) presents SciDTB, a discourse dependency treebank for scientific abstracts from ACL Anthology and implements transition-based and graph-based parsers on the dataset. Yi et al. (2021) adapts three Chinese discourse corpora to the dependency framework. We use these datasets to evaluate our model.

**Hierarchical Discourse Parsing** Hierarchical models are employed to improve the discourse dependency parsing performance for transition-based (Zhou and Feng, 2022), graph-based (Zhang et al., 2021a), and top-down (Kobayashi et al., 2020) models. These methods improve parsing results but they do not make full use of the parsing history, as is described in the introduction part. We store the predicted arcs and expand each EDU to a sub-tree and use a GAT-based encoder to merge information from EDUs in the sub-tree.

## 3 Methodology

Given a document composed of a sequence of EDUs $E = (e_1, e_2, \ldots, e_n)$, a discourse dependency parser is expected to predict a tree structure that represents the relationships between the EDUs. For each EDU $e_i \in E$, excluding the single root EDU of the document, a head EDU $h_i$ is selected from the set of EDUs $E$, and a relation $r_i$ is assigned from a pre-defined set of relations $R$ to describe the relationship between the EDU pair $(e_i, h_i)$. We adopt a pipelined approach (Wang et al., 2017) for the task. We construct a dependency tree structure for each document and then leverage this tree structure to predict the relations between EDUs.

### 3.1 Structure Prediction

In the structure prediction step, we utilize a hierarchical variant of transition systems. This section begins by introducing our hierarchical setting and transition system, followed by detailed implementations of our model.

### 3.1.1 Hierarchical Setting

Our method parses the document at three levels for RST-DT: EDU-to-sentence (E2S), sentence-to-paragraph (S2P), and paragraph-to-document (P2D). For SciDTB and CDTB, in which paragraph structures are absent, we refer to the first two levels as intra-sentence and inter-sentence. The parsing

process follows a bottom-up approach, where the sub-trees constructed in the preceding steps are utilized in the subsequent steps.

Figure 1 illustrates the parsing process employed for structure prediction. The document selected in SciDTB comprises five sentences with a total of 19 EDUs. Our approach involves initially constructing a sub-tree for each sentence, followed by merging these sub-trees to form a comprehensive tree representation.

We adopt the hierarchical setting for several reasons:

(1) The hierarchical structure reduces the average length of the sequence.

(2) Our approach aligns with the target of forming a tree structure.

(3) Observations of datasets: We define an EDU as an out-EDU if its head is in another paragraph. In RST-DT, 89.5% of paragraphs have only one out-EDU (root), and 86.1% of out-EDUs have another out-EDU as its head. These results are acceptable due to the relatively loose connections between paragraphs. The percentage is even higher for sentences. These observations encourage us to employ this simplified approach.

### 3.1.2 Transition System

At each level of structure prediction, we utilize an arc-eager transition system (Nivre, 2003) as our parsing approach. We employ a queue $\mathcal{Q}$ initialized with the input sequence and an initially empty stack $\mathcal{S}$ to store the EDUs. During the parsing process, we maintain a set $\mathcal{A}$ to record the acquired arcs derived from the transition history.

For each transition state $(\mathcal{Q}, \mathcal{S}, \mathcal{A})$, the discourse parser takes one of the following actions: $Shift$, $Reduce$, $LeftArc$ and $RightArc$. The $Shift$ action removes the first EDU in the queue $\mathcal{Q}$ and pushes it to the top of the stack $\mathcal{S}$. The $Reduce$ action pops the top EDU from the stack $\mathcal{S}$. The $LeftArc$ action adds an arc to the set $\mathcal{A}$, where the head is the first EDU in the queue $\mathcal{Q}$ and the dependent is the top EDU in the stack $\mathcal{S}$. After adding the arc, the parser performs the $Reduce$ action. The $RightArc$ action adds an arc to the set $\mathcal{A}$, where the head is the top EDU in the stack $\mathcal{S}$ and the dependent is the first EDU in the queue $\mathcal{Q}$. After adding the arc, the parser proceeds with the $Shift$ action. The parsing process continues until the queue $\mathcal{Q}$ is empty and only a single root EDU for the input sequence remains in the stack $\mathcal{S}$.

### 3.1.3 Our Model

In this section, we introduce how our transition system handles the current state.

(1) Select four source EDUs if available, including the top two EDUs from the stack and the first two EDUs from the queue.

(2) Construct a sub-tree for each source EDU with the arcs obtained in previous steps.

(3) Derive contextualized representations for each EDU in the sub-trees.

(4) Calculate representations for each sub-tree based on inner EDUs using GATs.

(5) Concatenate the sub-tree representations and employ a fully-connected layer for predicting the subsequent action.

We will also elucidate the implementation details of our model, specifically focusing on the methods for deriving representations for EDUs, sub-trees, and transition states..

**EDU Representations** To capture contextual representations of EDUs, we employ a pre-trained BERT (Devlin et al., 2019) model to encode each EDU within its corresponding context. For RST-DT, we consider the entire paragraph as the context for each EDU. For SciDTB and CDTB, we provide BERT with the sentence containing the EDU, as well as the preceding and succeeding sentences if available, to capture the contextual information. Sentences are delimited by $[SEP]$ tokens, while each sentence comprises EDUs separated by additional special tokens $[ES]$ representing "edu-sep". For each EDU, we compute its contextual representation by taking the token-level average of the corresponding EDU span.

**Sub-Tree Representations** In our hierarchical model, it is important to highlight that the set $\mathcal{A}$ not only stores arcs generated from preceding steps within the current EDU sequence but also captures lower-level parsing results, which should be utilized in upper-level parsing. For instance, in inter-sentence parsing, even though only the root EDUs from the intra-sentence sub-trees are used as input for the model, the sub-trees can be reconstructed using the intra-sentence arcs contained within $\mathcal{A}$.

Given a source EDU, we expand it into two distinct types of sub-trees: the **Star Graph** and the **Full Graph**. The methods to construct the sub-trees and calculate the sub-tree representations will be introduced in detail in section 3.1.4.

**Transition State Representations** For each transition state $(\mathcal{Q}, \mathcal{S}, \mathcal{A})$, we consider the top two EDUs in the stack $\mathcal{S}$ and the first two EDUs in the queue $\mathcal{Q}$ as source EDUs. We expand each source EDU into a sub-tree and calculate the sub-tree representation as we mentioned above. Finally, we obtain representations for the transition state by concatenating the representations of the four sub-trees. To predict the transition action, we employ a fully-connected layer.

### 3.1.4 Sub-Tree Construction

In this section, we present two distinct methods for expanding a source EDU into a sub-tree: the **Star Graph** and the **Full Graph**.

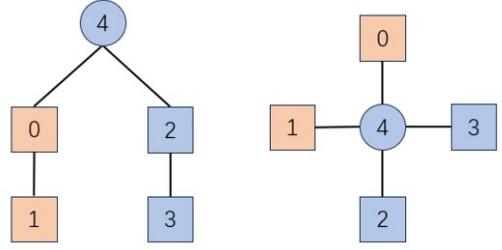

**Full Graph for EDU 4    Star Graph for EDU 4**

Figure 2: Full-Graph and Star-Graph for the source EDU 4. The source EDU is depicted with circular borders, whereas the expanded EDUs are represented with square borders. The graphs are constructed based on arcs stored in the above-mentioned transition states after three steps. The arc between EDU 4 and EDU 0 comes from the current inter-sentence sequence, while the others come from intra-sentence sequences.

**Star-Graph Model** A single EDU may not contain all the information we need and EDUs linked to it in precious steps may also convey important information. Therefore, we take these EDUs into consideration. Given the source EDU $e_s$ and its corresponding sub-tree $T$, we construct a star graph where $e_s$ is directly connected to all the other EDUs within $T$, as illustrated in Figure 2. Subsequently, we calculate a weighted average $V(T)$ over the EDUs in $T$ to represent the sub-tree.

$$V(T) = \sum_{e \in T} w(e)V(e) \qquad (1)$$

Here $V(e)$ is the representation of EDU $e$. The weights $w(\cdot)$ are determined by the similarity between each EDU and the source EDU $e_s$. We sim-

ply take the inner product as the similarity score.

$$w(e) = \frac{exp(sim(V(e), V(e_s)))}{\sum_{e_k \in T} exp(sim(V(e_k), V(e_s)))} \quad (2)$$

**Full-Graph Model** The star graph connects all EDUs in the sub-tree $T$ directly to the source EDU $e_s$, destructing the original structure of the sub-tree. We argue that the structure can also convey valuable information aside from the EDUs. For the full graph, we retain the complete structure of $T$. We view the sub-tree $T$ as an undirected graph, where each node corresponds to an EDU. We leverage GATs (Veličković et al., 2018) to incorporate neighboring information into the source EDU $e_s$.

In each graph attention layer, we conduct self-attention on node representations to calculate $a_{ij}$, which quantifies the significance of $e_j$ to $e_i$.

$$a_{ij} = \frac{exp(s_{ij})}{\sum_{k:e_k \in \mathcal{N}(e_i)} exp(s_{ik})} \quad (3)$$

The attention weight is applicable only when $e_j \in \mathcal{N}(e_i)$. Here $\mathcal{N}(e_i)$ denotes the set of first-order neighbors of $e_i$, which includes $e_i$ itself. The similarity score $s_{ij}$ is defined as

$$s_{ij} = LeakyReLU\left(\alpha^\top[Wv_i \| Wv_j]\right) \quad (4)$$

Here LeakyReLU is defined as follows and we set $\lambda = 0.2$.

$$LeakyReLU(x) = \begin{cases} x & , x \geq 0 \\ \lambda x & , x < 0 \end{cases}$$

$v_i, v_j$ are representations for $e_i, e_j$, and the matrix $W$ denotes a linear projection applied to the nodes. $W \in \mathbf{R}^{d_{out} \times d_{in}}$ and $\alpha \in \mathbf{R}^{2d_{out}}$ are learnable parameters. $d_{in}$ refers to the dimension of EDU representations and $\|$ is the concatenation operation. Then the representations for $e_i$ can be updated as follows:

$$v_i' = ELU\left(\Sigma_{j:e_j \in \mathcal{N}(e_i)} a_{ij} Wv_j\right) \quad (5)$$

Here ELU is defined as follows and we set $\lambda = 1$.

$$ELU(x) = \begin{cases} x & , x \geq 0 \\ \lambda(e^x - 1) & , x < 0 \end{cases}$$

According to Vaswani et al. (2017), multi-head attention enhances performance by simultaneously capturing information from different representation subspaces. Therefore we employ multi-head attention in our model. To merge information from multi-hop neighbors, we utilize multiple graph attention layers. This allows us to incorporate knowledge from distant nodes in the graph. We also apply residual connections and layer normalization techniques to further enhance the model performance. We extract the representations of the source EDU in the final layer as the sub-tree representations.

## 3.2 Relation Recognition

After the structure prediction step, we generate a dependency tree that includes nodes representing EDUs and edges representing relations between them. For the relation recognition step, we fine-tune two distinct BERT models for the nodes and edges within the tree structure. Subsequently we employ edge-featured GATs (Chen and Chen, 2021) to derive structure-aware contextualized representations for EDU pairs.

**NodeBERT** Given a constructed discourse dependency tree, we employ the same method described in section 3.1.3 to obtain representations for each EDU. For each EDU pair connected by an edge $(e_i, h_i)$, we concatenate their respective representations and utilize a fully-connected layer to predict their relation. We fine-tune the NodeBERT model with gold dependency tree structures in the training set.

**EdgeBERT** Similar to Zhou and Feng (2022), for each edge $(e_i, h_i)$ in the dependency tree, we concatenate the two EDUs in the order they appear in the original sequence. The concatenated sequence is then fed into a BERT model. Subsequently, the embedding of the special token $[CLS]$ is employed to predict the relation $r_i$ using a fully-connected layer. We also fine-tune the EdgeBERT model with gold dependency tree structures in the training set.

**Node+Edge Model** After the fine-tuning step, we utilize NodeBERT to encode each node and EdgeBERT to encode each edge. For each edge $(e_i, h_i)$ in the dependency tree, We concatenate the representations of the edge with those of the corresponding two nodes and predict the relation using a fully-connected layer.

**EGAT Model** We view each EDU as a node in the graph and head-dependant pairs as edges in the graph. Besides the edge itself and its corresponding nodes, the other nodes and edges may also contain important information for predicting the relation

for the edge. Based on the representations for nodes and edges, we apply EGATs (Chen and Chen, 2021) to the graph. EGAT is similar to GAT introduced in section 3.1.4, except for the following rules:

(1) Edge features are employed to compute attention scores between nodes. We adapt the similarity score (Eq. 4) between node $i$ and $j$ as follows:

$$s_{ij} = LeakyReLU(\alpha^\top [W_n v_i \| W_n v_j \| W_e u_{ij}]) \tag{6}$$

Here, $\alpha \in \mathbf{R}^{3d_{out}}$, $W_n, W_e \in \mathbf{R}^{d_{out} \times d_{in}}$ denote linear projections for nodes and edges, respectively. The representations for nodes and edges are denoted as $v$ and $u$.

(2) In addition to updating node representations, we construct a dual graph to update edge representations. Each node in the dual graph $G'$ corresponds to an edge in the original graph $G$. Two nodes in $G'$ are connected if and only if their corresponding edges in $G$ share a common node. Consequently, we can update node representations in $G'$ (as well as edge representations in the original graph $G$) using the same method as described in (1).

(3) Self-loops are added for both $G$ and $G'$. In the original graph $G$, each node is assigned a self-loop that aggregates the features of all edges connected to it by taking their average. In the dual graph $G'$, each node possesses a self-loop with zero-valued features. More details can be found in Chen and Chen (2021).

After applying multiple EGAT layers to update the representations of nodes and edges, we characterize each EDU pair $(e_i, h_i)$ by concatenating the representations of the two EDUs and the edge connecting them. Finally, we utilize a fully connected layer to predict the relation.

## 4 Experiments

### 4.1 Dataset and Evaluation Metric

We evaluate our model on three datasets: RST-DT (Carlson et al., 2001), SciDTB (Yang and Li, 2018) and CDTB (Li et al., 2014b). RST-DT is a dataset that consists of Wall Street Journal articles. SciDTB is a dataset comprising scientific abstracts from the ACL Anthology. CDTB is a dataset containing Chinese newswire articles. To convert the constituency trees into dependency trees, we follow the approach proposed by Li et al. (2014a) for RST-DT and the method introduced by Yi et al. (2021) for CDTB. More detailed information about these datasets is provided in Table 1.

|  | RST-DT | SciDTB | CDTB |
|---|---|---|---|
| #doc (train) | 312 | 743 | 1599 |
| #doc (dev) | 30 | 154 | 350 |
| #doc (test) | 38 | 152 | 374 |
| #rel (coarse) | 19 | 17 | 18 |
| #rel (fine) | 111 | 26 | – |
| #para/docu | 10.03 | 1 | 1 |
| #sent/para | 2.01 | 5.33 | 2.03 |
| #edu/sent | 2.76 | 2.63 | 2.23 |
| #token/edu | 10.85 | 11.10 | 22.76 |

Table 1: Detailed information about the datasets, including the dataset split, the number of coarse-grained and fine-grained relation types and the average number of paragraphs in documents, sentences in paragraphs, EDUs in sentences and BERT tokens in EDUs.

We employ the unlabeled attachment score (UAS) and labeled attachment score (LAS) as evaluation metrics to assess the performance of our model. The UAS measures the percentage of EDUs in which the model correctly predicts the head, while the LAS measures the percentage of EDUs where the model correctly predicts both the head and the relation.

### 4.2 Baselines

Since there are only a few works in discourse dependency parsing, we only compare our model with the state-of-the-art transition-based and graph-based models. For fair comparison, the pre-trained language models used are *bert-base-uncased* for English and *bert-base-chinese* for Chinese.

(1) **Zhou22** : Zhou and Feng (2022) proposes a hierarchical transition-based model for structure prediction and uses a sequence labeling model to predict discourse relations.

(2) **Zhang21** : Zhang et al. (2021a) utilizes a hierarchical Eisner model with neural CRF autoencoders to simultaneously predict heads and relations. For fair comparison, we consider the fully-supervised version of their approach.

### 4.3 Hyper-Parameters

Here we present the value of several important hyper-parameters:

(1) The dimension of token, EDU and sub-tree representations is set to 768. In Equation 4 and 6, we set $d_{in}$ to 768 and $d_{out}$ to $768/H$, where $H$ represents the number of attention heads.

(2) For GATs, we explore different settings for the number of layers ($L$) chosen from $\{1, 2, 3, 5\}$, the number of attention heads ($H$) selected from

| stage | L | H | bs | lr | epochs |
|---|---|---|---|---|---|
| structure(E2S) | 2 | 3 | 16 | 2e-5 | 5 |
| structure(S2P&P2D) | 3 | 6 | 8 | 2e-5 | 5 |
| relation | 3 | 3 | 32 | * | 10 |

Table 2: Hyper-parameters in different stages of our model. The cell with an asterisk: lr=1e-6 for BERT parameters and 1e-4 for the others

$\{1, 3, 6, 12\}$, batch size ($bs$) from $\{8, 16, 32\}$, as well as the learning rate ($lr$) and the number of training epochs.

We list the values in different stages in Table 2.

### 4.4 Main Results

We report the results of our model with Full Graph in section 3.1.4 for structure prediction and EGATs in section 3.2 for relation recognition, and compare our model with the baselines mentioned above.

We conduct a comparison between our model and **Zhou22** on SciDTB (with 26 fine-grained relations) and CDTB (with 18 relations). The corresponding results are presented in Table 3. These results demonstrate that our model surpasses the most recent transition-based model for discourse dependency parsing. The increase in UAS (3.20% for SciDTB and 2.60% for CDTB) highlights the advantages of our GAT-based sub-tree encoder, as our model incorporates GATs to capture sub-tree representations for transition states instead of solely considering isolated EDUs. The improvement in LAS (4.89% for SciDTB and 4.98% for CDTB) indicates the superiority of our EGAT-based relation classifier over sequence labeling models. By comparing the mean and standard deviation of UAS and LAS score, we can see that our model outperforms **Zhou22** by a large margin in both structure construction and relation recognition, even when compared with the higher reported results, indicating the superiority of our GAT-based model. We will provide a comprehensive comparison between our model and **Zhou22** in section 4.5.

Besides, we compare our model with **Zhang21** on RST-DT (with 19 coarse-grained relations) and SciDTB (with 17 coarse-grained relations). As shown in Table 4, our model exhibits significant improvements over the latest graph-based discourse dependency parsing model, even when applied to longer documents in the RST-DT dataset. This finding suggests that our model effectively handles distant EDU pairs within a lengthy context.

|  |  | UAS | LAS |
|---|---|---|---|
| **SciDTB** | **Zhou22-rep** | 79.3 | 65.0 |
|  | **Zhou22-run** | 78.08 ± 1.06 | 62.42 ± 0.72 |
|  | Our model | **81.28 ± 0.34** | **67.31 ± 0.44** |
| **CDTB** | **Zhou22-rep** | 82.2 | 64.8 |
|  | **Zhou22-run** | 81.76 ± 1.18 | 62.63 ± 0.87 |
|  | Our model | **84.36 ± 0.05** | **67.61 ± 0.36** |

Table 3: Discourse dependency parsing results on SciDTB (with 26 fine-grained relations) and CDTB (with 18 relations). We run both **Zhou22**(**Zhou22-run**) and our model for six times, and we report the mean and standard deviation of the results for each model. Our re-implemented results of **Zhou22** is lower than those reported in the paper, so we also list the reported results here(**Zhou22-rep**).

|  | SciDTB (c) | | RST-DT (c) | |
|---|---|---|---|---|
|  | UAS | LAS | UAS | LAS |
| **Zhang21** | 79.1 | 65.0 | 70.2 | 51.8 |
| Our model | **81.28** | **71.23** | **71.41** | **55.64** |

Table 4: Discourse dependency parsing results on SciDTB (with 17 coarse-grained relations) and RST-DT (with 19 coarse-grained relations). We use the reported results in **Zhang21**. For our model, we take the average results of three runs.

### 4.5 Detailed Results

Since our model follows a similar pipeline procedure as **Zhou22**, we present the results of each step and conduct a thorough comparison on SciDTB.

**Discourse Structure Prediction** We partition the EDUs, excluding the root EDU of each document, into two subsets: $E_{intra}$ and $E_{inter}$. An EDU is categorized as part of $E_{intra}$ if it belongs to the same sentence as its head EDU, and as part of $E_{inter}$ otherwise. We evaluate model performance on these two subsets.

We conduct experiments at intra-sentence and inter-sentence levels and define the **Intra** score as the percentage of EDUs in $E_{intra}$ for which the head EDU is correctly predicted. **Inter_p** and **Inter_g** are similar scores for $E_{inter}$ based on predicted and gold intra-sentence structure separately.

We evaluate three models using these scores, all of which are transition-based models. The primary difference among them lies in the representations of sub-trees. **Zhou22** exclusively utilizes the source EDU for representation, therefore we refer to this approach as the **No-Graph** model. In contrast, our two graph models expand the source EDU to encompass a sub-tree by incorporating the

| | | |
|---|---|---|
| **Intra** | No-Graph | 87.45 ± 0.86 |
| | Ours (Star Graph) | 88.85 ± 0.70 |
| | Ours (Full Graph) | **89.09 ± 0.35** |
| **Inter_p** | No-Graph | 63.21 ± 1.49 |
| | Ours (Star Graph) | 65.36 ± 1.28 |
| | Ours (Full Graph) | **67.81 ± 0.88** |
| **Inter_g** | No-Graph | 66.54 ± 1.30 |
| | Ours (Star Graph) | 68.73 ± 1.13 |
| | Ours (Full Graph) | **71.16 ± 0.66** |

Table 5: Structure prediction results on SciDTB. To obtain results for the **No-Graph** model, we re-implement the approach described in **Zhou22**, for some of the scores defined here are not reported in the original paper. We present here the mean and standard deviation of results for six runs.

arcs obtained during preceding parsing steps. Our **Star-Graph** model constructs a star graph by directly connecting all other EDUs in the sub-tree to the source EDU and our **Full-Graph** model performs GATs on the original sub-tree structure. We provide detailed explanations in section 3.1.4. The results are presented in Table 5. By comparing the mean and standard deviation of the results, we can draw the following conclusions.

For intra-sentence parsing, both our **Star-Graph** and **Full-Graph** models surpass the performance of the **No-Graph** model, emphasizing the need to integrate information from all EDUs within the sub-tree. However, the **Full-Graph** model demonstrates only marginal improvements compared to the **Star-Graph** model. This observation can be attributed to the relatively small number of EDUs in each sentence and the simple sentence structures, suggesting that a simplified star graph is sufficient for merging information between EDUs within a sentence.

For inter-sentence parsing, our **Star-Graph** and **Full-Graph** models outperform the **No-Graph** model by a larger margin. This superiority arises from their ability to leverage arcs obtained from intra-sentence parsing to enhance inter-sentence parsing. Unlike results in intra-sentence parsing, the **Full-Graph** model significantly outperforms the **Star-Graph** model. This observation highlights the critical role of the graph structure in comprehending text spans with a larger number of EDUs and more complex structures.

**Discourse Relation Recognition** We evaluate discourse relation recognition models based on gold dependency trees. We define **Intra_g**, **In-**

| | Intra_g | Inter_g | LAS_g |
|---|---|---|---|
| Seq-label | 82.4 | 62.2 | 77.4 |
| Ours (Node+Edge) | **83.5** | 64.4 | 78.6 |
| Ours (EGAT) | 83.3 | **66.2** | **79.0** |

Table 6: Relation recognition results on SciDTB with fine-grained relations and gold tree structure. We use the results reported in **Zhou22** for the **Seq-label** model, and for our model, we report the average results of six runs.

**ter_g** and **LAS_g** as the accuracy score for relation recognition in $E_{intra}$, $E_{inter}$ and all the EDUs. **Zhou22** treats discourse relation recognition as a sequence labeling task, and we refer to it as **Seq-label**. We compare it with our **Node+Edge** and **EGAT** models introduced in section 3.2. The results are presented in Table 6.

**Node+Edge** achieves superior performance compared to **Seq-label** primarily due to the integration of two fine-tuned BERT models. **NodeBERT** captures contextual representations for each EDU, while **EdgeBERT** facilitates direct interactions between EDU pairs. The **Seq-label** model contains only the latter. **EGAT** outperforms **Seq-label** at inter-sentence level, further emphasizing the significance of graph structures in handling complex text spans.

## 5 Analysis

It is evident that discourse dependency parsing performance declines when applied to lengthy text spans with complex structures. In this section, we conduct experiments on SciDTB to show the capability of our model in handling complex structures. Additionally, we perform experiments on RST-DT to showcase the superiority of our model in dealing with long text spans.

### 5.1 Results over complex structures

We perform inter-sentence experiments on SciDTB using gold intra-sentence structures. We divide the root EDUs of sentences into subsets based on the number of EDUs in the sentence and the sentence containing the head EDU.

The results are presented in Table 7. For most of the subsets, both the **Star-Graph** and **Full-Graph** models outperform the **No-Graph** model, indicating the necessity of merging information from all the elementary discourse units (EDUs) within a sub-tree. Furthermore, when comparing the **Full-Graph** model with the **Star-Graph** model, it is ev-

| N_edu | No-Graph | Star-Graph | Full-Graph |
|---|---|---|---|
| 2 (24) | **66.67** | 65.28 | 64.58 (-0.70) |
| 3 (98) | 58.50 | 62.24 | **62.41 (+0.17)** |
| 4 (148) | 65.99 | 64.98 | **66.55 (+0.57)** |
| 5 (157) | 63.06 | 63.27 | **65.50 (+2.23)** |
| 6 (108) | 66.36 | 68.21 | **72.53 (+4.32)** |
| 7 (69) | 58.94 | 59.66 | **64.01 (+4.35)** |
| 8 (30) | 64.44 | 63.89 | **73.33 (+9.44)** |
| >8 (32) | 58.31 | 58.84 | **62.50 (+3.66)** |

Table 7: Inter-sentence level structure prediction results in different subsets of SciDTB. The numbers in the brackets in the first column represent the size of each subset and the numbers in the brackets in the last column indicate the performance gain **Full-Graph** achieves compared to **Star-Graph**.

ident that the **Full-Graph** model achieves greater improvements when there are more EDUs in the corresponding sentences. When sentences contain a higher number of Elementary Discourse Units (EDUs), they often exhibit more complex structures. This finding demonstrates the capability of our GAT-based sub-tree representations to handle text spans with complex structures.

## 5.2 Results over long text spans

We conduct experiments with our model on RST-DT, a dataset consisting of documents with an average of 10.03 paragraphs. We evaluate the results of P2D-level parsing. For the score **P2D_g**, we use gold E2S and S2P structures and for **P2D_p**, we utilize E2S and S2P structures predicted by our **Full-Graph** model.

We present the performance of three approaches in Table 8: **No-Graph**, **Star-Graph**, and **Full-Graph**. The **No-Graph** approach is a modified version of our model where the sub-tree representation utilizes only the source EDU. Based on the results, it is evident that our **Full-Graph** model surpasses the performance of the **No-Graph** model, highlighting the superiority of our GAT-based sub-tree encoder. The **Star-Graph** model achieves a lower score, possibly due to the destruction of the original sub-tree structure. Since there are numerous EDUs in a sub-tree during S2P-level parsing, when we simplify the sub-tree to a star graph, the information carried by nearby nodes may be contaminated by distant nodes.

For further study on this topic, we partition the root EDUs of paragraphs into subsets of approximately equal size, taking into account their distance from the head EDU. The distance between

| | No-Graph | Star-Graph | Full-Graph |
|---|---|---|---|
| **P2D_p** | 25.6 | 22.4 | **30.0** |
| **P2D_g** | 36.3 | 33.3 | **41.1** |

Table 8: P2D-level structure prediction results on RST-DT.

| Dis_edu | No-Graph | Star-Graph | Full-Graph |
|---|---|---|---|
| 1~4 (130) | 45.4 | 28.5 | **50.0** |
| 5~8 (110) | 31.0 | 28.2 | **41.0** |
| 9~18 (106) | 22.6 | 20.8 | **35.9** |
| >18 (93) | 21.5 | **35.5** | 12.9 |

Table 9: P2D-level structure prediction results in different subsets of RST-DT. The numbers in the brackets in the first column represent the size of each subset.

two EDUs is measured by the number of EDUs between them, with a value of 1 for adjacent EDUs.

We compute the **P2D_g** scores for these subsets and present the results in Table 9. Based on the obtained results, we can draw the following conclusions: (1) Our **Full-Graph** model demonstrates superior performance compared to the **No-Graph** model across all subsets, except for the subset involving extremely long distances. Furthermore, this superiority becomes more pronounced as the subsets involve longer distances. These findings indicate that our **Full-Graph** model exhibits a remarkable capability to handle longer spans. (2) The **Star-Graph** model yields a lower score for adjacent EDU pairs but excels in the case of distant ones. This observation provides evidence that the destruction of the sub-tree structure indeed results in data contamination mentioned above. The effectiveness of our GAT-based sub-tree encoder is validated by these experiments.

## 6 Conclusion

In this paper, we enhance hierarchical transition systems by integrating a GAT-based sub-tree encoder, which enables the parser to leverage the parsing history more effectively and have a better understanding of the whole document. Additionally, we apply edge-featured GATs to predict relations based on the complete discourse structure, where two separate BERT models are fine-tuned to capture features for EDUs and EDU pairs, respectively. Experimental results demonstrate that our model outperforms existing methods, particularly when applied to long text spans with complex structures.

## Limitations

Since the sub-tree changes during the transition-based parsing process, we have to encode different sub-trees for each transition state, which is time-consuming. As a result, our structure prediction model requires more time compared to existing models. This issue becomes particularly prominent when dealing with long documents in RST-DT. In future works, We will design a more efficient graph structure for encoding sub-trees. Besides, when evaluating our model on CDTB, we encounter a significant issue of data imbalance, particularly in the stage of inter-sentence structure prediction, where $Shift$ and $RightArc$ comprise the majority of the actions. Consequently, our model has a tendency to produce fully right-branching trees. We leave the problem for future works.

## Acknowledgements

This work is supported by NSFC (62161160339). We would like to thank the anonymous reviewers for their helpful comments and suggestions.

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
