# OpenReview forum: "Learning Dynamic Representations for Discourse Dependency Parsing"
_EMNLP/2023/Conference — EMNLP 2023 Findings_

### Official Review · Reviewer_pLZG · 2023-07-29

**Typos Grammar Style And Presentation Improvements:** L046
**Soundness:** 3

**Excitement:**

3: Ambivalent: It has merits (e.g., it reports state-of-the-art results, the idea is nice), but there are key weaknesses (e.g., it describes incremental work), and it can significantly benefit from another round of revision. However, I won't object to accepting it if my co-reviewers champion it.

**Paper Topic And Main Contributions:**

This work proposes a hierarchical transition-based model for the task of discourse dependency parsing. The parsing procedure is performed with multiple levels, and the partial structures obtained from previous transitions are models with a GAT-based encoder to leverage parsing history. A similar GAT-based model is utilized to predict discourse relations. With evaluations on three datasets, the proposed models are shown to perform better than previous methods and other baselines.

**Questions For The Authors:**

- A: It seems that at each transition-step, the EDU representations are re-encoded with the updated structures? Is this necessary, and are there any incremental solutions the authors may have considered?
- B: Instead of the pipelined approach that separately predicts the structures and the labels, would it be beneficial to predict them jointly?
- C: Are there any specific reasons to adopt the arc-eager system? How about other transition systems, such as arc-standard?


**Reasons To Accept:**

- Modeling partial structures during the parsing procedure is an intuitive idea and it is nice to see that it works for the discourse-level parsing task.
- The proposed model is reported to obtain better results than previous methods.


**Reasons To Reject:**

- There are few baselines that are compared and ablation studies are lacking. Though there may be few works on this task, it would be nice to provide more analysis to illustrate which model designs bring the most improvements. Especially, “Zhou22” is also a hierarchical transition-based model and would be a potentially good start point to perform the ablations.
- Although mentioned in the Limitation section, there is no analysis on the efficiency of the proposed model. Encoding the sub-tree structures may bring large extra costs, which should be carefully considered to better apply the proposed model.


**Reproducibility:**

3: Could reproduce the results with some difficulty. The settings of parameters are underspecified or subjectively determined; the training/evaluation data are not widely available.

**Reviewer Confidence:**

3: Pretty sure, but there's a chance I missed something. Although I have a good feel for this area in general, I did not carefully check the paper's details, e.g., the math, experimental design, or novelty.

---

> ### Author Rebuttal · Authors · 2023-08-29
>
> Thanks for your time and helpful suggestions in our work.
>
> For the question of baselines, there are few works on the discourse dependency parsing task, and most early works do not use pre-trained language models. Therefore we only select two baselines, which are the SOTA transition-based and graph-based models. Our model and the two baselines employ the same PLM (BERT-base) for fair comparison.
>
> For the question of ablation study, it's important to note that 'No-Graph' in Table 4 maintains an identical structure to Zhou22, with only minor modifications in the encoding component.
> The results presented in Table 4 clearly demonstrate the advantages of incorporating EDUs within the sub-tree structure. Specifically, 'Star-Graph' outperforms 'No-Graph,' highlighting the benefits of including EDUs. Moreover, the introduction of a sub-tree structure further enhances performance, as evidenced by 'Full-Graph' surpassing 'Star-Graph’. Besides, Table 4 and 5 illustrate the performance of our model at different parsing levels, and Table 6, 7 and 8 show the performance of our model on long texts.
>
> For the question of efficiency, we admit that our model incurs significant computational costs. This becomes particularly noticeable when dealing with longer texts, especially during the P2D stage in the RST-DT dataset. However, it manages to achieve a much better performance in SciDTB and CDTB with an acceptable increase in processing time.
>
> When comparing our full-graph model to the no-graph version (similar to Zhou22, which uses the representation of a single root EDU) at both E2S and S2P levels on SciDTB and CDTB, the time requirements for each transition step during training are as follows:
>
> E2S Level:
>
> SciDTB: 0.063s (full-graph) vs. 0.051s (no-graph)
>
> CDTB: 0.067s (full-graph) vs. 0.049s (no-graph)
>
> S2P Level:
>
> SciDTB: 0.206s (full-graph) vs.0.063s (no-graph)
>
> CDTB: 0.195s (full-graph) vs. 0.056s (no-graph)
>
> Notably, at the E2S level, both versions demand a similar amount of time, making both of them efficient. Even at the S2P level, where the processing time increases, it remains within tolerable limits. Consequently, our method is well-suited for a wide range of applications, except for cases involving exceptionally long texts where efficiency concerns may arise.
>
> For Question A, the EDU representations are re-encoded after each step. We believe this is beneficial to update the EDU representation with different sub-tree structures although it is time costly. In our future research, we plan to explore the possibility of employing an approximate method for updating EDU representations within a sub-tree. This method would involve merging the representations of two connected sub-trees rather than completely re-encoding them from scratch.
>
> For Question B, the datasets for the discourse dependency parsing task are relatively small. The available data within each class may be too sparse if we attempt to predict both the structure and relations jointly.
>
> Regarding Question C, we think the arc-standard system may be inferior to the arc-eager system, because the latter permits an EDU to establish a connection with its head earlier in the parsing process, bypassing the need to wait for all of its children to be parsed, which may be helpful for long texts. We will perform experiments on this argument in our future work. Thank you for your advice.

---

### Official Review · Reviewer_m6QC · 2023-08-04

**Soundness:** 4

**Excitement:**

4: Strong: This paper deepens the understanding of some phenomenon or lowers the barriers to an existing research direction.

**Missing References:**

None

**Paper Topic And Main Contributions:**

The paper used a GAT-based encoder to dynamically represent sub-trees constructed in transition history. Further, the paper proposed a novel parsing model based on the representations produced by the GAT-based encoder. The experimental results showed the superiority of the proposed model on multiple text corpora.

**Questions For The Authors:**

Nothing particular, but comments on the weakness above could be helpful.

**Reasons To Accept:**

The model proposed in the paper is elaborately described. Empirical results and the corresponding analysis are sufficient to illustrate the superiority of the proposed model.

**Reasons To Reject:**

The authors claimed the proposed model achieved a significant improvement without providing any statistical evidence. The authors should provide several proper statistics to quantify the variability in the performance of their proposed model.

**Reproducibility:**

4: Could mostly reproduce the results, but there may be some variation because of sample variance or minor variations in their interpretation of the protocol or method.

**Reviewer Confidence:**

3: Pretty sure, but there's a chance I missed something. Although I have a good feel for this area in general, I did not carefully check the paper's details, e.g., the math, experimental design, or novelty.

**Typos Grammar Style And Presentation Improvements:**

line 023: resarch -> research
line 046: levles -> levels
line 161: make full of -> make full use of

---

> ### Author Rebuttal · Authors · 2023-08-29
>
> Thank you for the useful suggestions which will definitely improve our work.
>
> Regarding the lack of statistical evidence, most of the results reported in our paper are the average of six separate runs. We provide both the mean values and their corresponding standard deviations for the experiments shown in Table 2 and Table 4 here:
>
> Table 2 :  Discourse dependency parsing results on SciDTB (with 26 fine-grained relations) and CDTB (with 18 relations) :
>
> |  model   | SciDTB_UAS |   SciDTB_LAS |  CDTB_UAS |  CDTB_LAS |
>
> | Zhou22	| 78.08 ± 1.06 | 62.42 ± 0.72 | 81.76 ± 1.18 | 62.63 ± 0.87 |
>
> | Ours       | 81.28 ± 0.34 | 67.31 ± 0.44 | 84.36 ± 0.05 | 67.61 ± 0.36 |
>
> Table 4 : Structure prediction results on SciDTB :
>
> | model          | Intra sentence   |   inter_p  |     inter_g |
>
> | No-Graph | 87.45 ± 0.86 | 63.21 ± 1.49 | 66.54 ± 1.30 |
>
> | Star-Graph | 88.85 ± 0.70 | 65.36 ± 1.28 | 68.73 ± 1.13 |
>
> | Full-Graph | 89.09 ± 0.35 | 67.81 ± 0.88 | 71.16 ± 0.66 |
>
> In Table 2, it becomes evident that our model exhibits a significant performance advantage over Zhou22. Furthermore, Table 4 reveals remarkable distinctions between No-Graph and Star-Graph, as well as between Star-Graph and Full-Graph, particularly at the inter-sentence level.
>
> We will conduct a significance test for the remaining experiments in a forthcoming version of this study.

---

### Official Review · Reviewer_Hmaa · 2023-08-06

**Soundness:** 3

**Excitement:**

3: Ambivalent: It has merits (e.g., it reports state-of-the-art results, the idea is nice), but there are key weaknesses (e.g., it describes incremental work), and it can significantly benefit from another round of revision. However, I won't object to accepting it if my co-reviewers champion it.

**Paper Topic And Main Contributions:**

This paper presents a hierarchical prediction method for discourse dependency parsing of documents. Intuitively, a sentence contains a set of discourse units that have discourse relations between them; a paragraph contains a set of sentences that have discourse relations between them; a text contains a set of paragraphs that have discourse relations between them. The key method proposed here is to use a rich learned representation of substructures when predicting relations at higher levels, rather than simple heuristic relations used in prior work. Experiments on multiple standard discourse parsing datasets show that this method outperforms previously proposed methods, and deeper analysis illustrates how specific design decisions contribute to this superiority.

**Questions For The Authors:**

A: The authors acknowledge that discourse dependency parsing "receives relatively less attention" than discourse constituency parsing. What is the reason for that? Is there a reason why discourse dependency parsing is still attractive?

B: As the authors acknowledge, the results in Table 8 show that taking graph structure into account gives increasing improvements for long-distance connections between EDUs up until the final category of EDUs with distance >18 between them. Is there a hypothesis for what is happening in this category? Is the data noisy or does the model hit some fundamental scaling issues?

C: Are relationships between specific sentences in different paragraphs modelled in this approach?

**Reasons To Accept:**

The technical idea makes sense and the experimental results give strong evidence that it works well. Detailed analysis of the results corroborate the hypothesis that the hierarchical prediction method primarily gives benefits over discourse structures that are larger and more complex. These results will contribute to our understanding of discourse dependency parsing and give ideas to other researchers and practitioners.

**Reasons To Reject:**

While the technical content of the work appears solid, there are clarity issues which makes it hard for the reader to have confidence they fully understand the contribution, or would be able to replicate the work if they wish to:

- There is no discussion of hyperparameters, which will make it hard to replicate the work. E.g. the dimensions of parameter matrices are not given; authors state that they apply "multiple EGAT layers to update the representations of nodes and edges" but don't say how many is meant by "multiple".

- The algorithm used is complex and has many steps. This means that it has to be explained very clearly. I felt the explanation was not as clear as it could be, even after reading through Section 3 multiple times - I'm sure that if I tried to replicate I would get some details wrong. An appendix containing a fully-worked step-by-step example would be a helpful addition. Some other suggestions:

(a) Make it clear whether "Transition State Representations" and "EDU Representations" as described in Section 3.1.2 are relevant only to Section 3.1.2 (as the paper structure suggests) or also Section 3.1.1 (which I am guessing is the case)

(b) State explicitly the decision function that is used to classify the next parser action: what are its parameters and arguments?

- A commitment to release code would also help with reproducibility


**Reproducibility:**

3: Could reproduce the results with some difficulty. The settings of parameters are underspecified or subjectively determined; the training/evaluation data are not widely available.

**Reviewer Confidence:**

2: Willing to defend my evaluation, but it is fairly likely that I missed some details, didn't understand some central points, or can't be sure about the novelty of the work.

**Typos Grammar Style And Presentation Improvements:**

p.1, line 44: space in "distance ."

p.1, line 45: typo "levles"

p.3, line 158: space in "2022) ,"

p. 4 "LeakyReLU", "ELU", v_i' are not defined; please give definitions or citations

p.8, line 571: I think "D2P" should be "P2D" as used elsewhere in the paper

---

> ### Author Rebuttal · Authors · 2023-08-29
>
> Thank you for your efforts and time in our paper, which we believe is crucial to improve our work.
>
> For hyperparameters, we will give a detailed description in the future version. Here we present several important ones:
>
> (1)	The dimension of token, EDU, and sub-tree embeddings is set to 768. In Equation (4), we set d_in to 768 and d_out to 768 / H, where H represents the number of attention heads.
>
> (2)	For GATs, we explore different settings for the number of layers (L) chosen from {1, 2, 3, 5}, the number of attention heads (H) selected from {1, 3, 6, 12}, batch size (bs) from {8, 16, 32}, as well as the learning rate and the number of training epochs.
>
> a)	In the E2S structure stage: L=2, H=3, bs=16, lr = 2e-5, epochs=5
>
> b)	In the S2P and P2D structure stage: L=3, H=6, bs=8, lr=2e-5, epochs=5
>
> c)	In the relation prediction stage (EGATs): L=3, H=3, bs=32, lr=1e-6 for BERT parameters and 1e-4 for the others, epochs=10.
>
> For Question (a), section 3.1.1 and 3.1.2 are closely related and can be merged into one section. We put "Transition State Representations" and "EDU Representations" in section 3.1.2 because we consider them as part of the sub-tree encoder. Sorry for the confusion. We will clarify in the revised version.
>
> For question (b), our approach involves the following steps:
>
> (1)	select the top two EDUs from the stack and the first two EDUs from the queue.
>
> (2)	construct a sub-tree for each source EDU with the arcs obtained in previous steps
>
> (3)	derive representations for source EDUs based on their respective sub-trees using GATs.
>
> (4)	concatenate the representations and employ a fully-connected layer for predicting the subsequent action.
>
> We will add a detailed algorithm and release our code for further study.
>
> For Question A, discourse constituency parsing is more popular because it can capture the structure of a discourse more accurately, primarily attributed to the presence of interior nodes representing text spans. On the other hand, discourse dependency parsing can be helpful for downstream tasks, especially information extraction tasks. For instance, it becomes particularly valuable when handling implicit temporal relation recognition tasks, as the length of the path between EDUs may play a crucial role in such scenarios.
>
> For Question B, the star-graph model demonstrates effective performance on extremely long texts. This success can be attributed to the direct connection of each EDU with the source EDU, enabling the extraction of comprehensive global information. However, the full-graph model performs even worse than no-graph, possibly due to
>
> (1) Limited Training Data: The long texts in the dataset are insufficient to train the model.
>
> (2) Increased Complexity: As the distance between EDUs increases, the model must make a greater number of accurate parsing decisions.
>
> (3) Information Transfer Challenges: The longest discourse contains 303 EDUs, 162 sentences, 45 paragraphs. Simple GATs may face difficulties in effectively transferring information within a complex tree with too many nodes.
>
> For Question C, we do not consider the relationships between specific sentences in different paragraphs except for the root sentences. We follow the hierarchical structure in which a sub-tree is constructed for each paragraph and they are subsequently merged to form a complete tree. In this arrangement, connections between paragraphs only exist between root EDUs of each paragraph. We adopt this approach for several reasons:
>
> (1)	The hierarchical structure reduces the average length of the sequence.
>
> (2)	Our approach aligns with the target of forming a tree structure.
>
> (3)	Observations: We define an EDU as an out-EDU if its head is in another paragraph. 89.5% of paragraphs have only one out-EDU (root), and 86.1% of out-EDUs have another out-EDU as its head. These results are acceptable due to the relatively loose connections between paragraphs. These observations encourage us to employ this simplified approach.
>
> In our future work, we will consider employing more complex structure. For example, we may allow data transferring between all pairs of sentences using a complex graph structure, as opposed to the current tree-based approach.
>
> Thanks for pointing out those typos, which we will fix in the later version.

---

### Meta-Review · Area_Chair_2n6e · 2023-09-19

**Recommendation:** 3

**Metareview:**

This paper presents a method relying on EGAT (edge graph attention networks) for hierarchical discourse dependency parsing. Several experiments and comparisons are reported showing that the method outperforms comparable methods on several standard corpora. Further analysis shows that the method is good for complex longer discourse dependencies/structures. All the reviewers agree on these.

The paper contributes to the field because there's relatively few works on discourse dependency parsing in general, or models that . Besides, the experimentation is good, sound and broad. However, there's an incremental feeling to the main ideas in that they basically adapt some existing methodology to an existing problem.

---

### Decision · Program_Chairs · 2023-10-07

**Decision:**

Accept-Findings

**Comment:**

This paper presents a method relying on EGAT (edge graph attention networks) for hierarchical discourse dependency parsing. Several experiments and comparisons are reported showing that the method outperforms comparable methods on several standard corpora. Further analysis shows that the method is good for complex longer discourse dependencies/structures. All the reviewers agree on these.

The paper contributes to the field because there's relatively few works on discourse dependency parsing in general, or models that . Besides, the experimentation is good, sound and broad. However, there's an incremental feeling to the main ideas in that they basically adapt some existing methodology to an existing problem.